# Histone Extraction from Human Articular Cartilage for the Study of Epigenetic Regulation in Osteoarthritis

**DOI:** 10.3390/ijms23063355

**Published:** 2022-03-20

**Authors:** Carmen Núñez-Carro, Margarita Blanco-Blanco, Tatiana Montoya, Karla M. Villagrán-Andrade, Tamara Hermida-Gómez, Francisco J. Blanco, María C. de Andrés

**Affiliations:** 1Unidad de Epigenética, Grupo de Investigación en Reumatología (GIR), Instituto de Investigación Biomédica de A Coruña (INIBIC), Complexo Hospitalario Universitario, de A Coruña (CHUAC), Sergas, 15006 A Coruña, Spain; carmen.nunez.carro@sergas.es (C.N.-C.); margarita.blanco.blanco@sergas.es (M.B.-B.); kariuxva@hotmail.com (K.M.V.-A.); tamara.hermida.gomez@sergas.es (T.H.-G.); fblagar@sergas.es (F.J.B.); 2Department of Pharmacology, Faculty of Pharmacy, Universidad de Sevilla, 41012 Sevilla, Spain; tmontoya@us.es; 3Centro de Investigación Biomédica en Red de Bioingeniería, Biomateriales y Nanomedicina (CIBER-BBN), Av. Monforte de Lemos, 3-5, Pabellón 11, 28029 Madrid, Spain; 4Grupo de Investigación en Reumatología y Salud, Centro de Investigaciones Científicas Avanzadas (CICA), Departamento de Fisioterapia, Facultad de Fisioterapia, Universidade da Coruña, Medicina y Ciencias Biomédica, 15006 A Coruña, Spain

**Keywords:** histone, chondrocytes, cartilage, methods, epigenetics, osteoarthritis, inflammation

## Abstract

Osteoarthritis (OA) is a chronic disease that affects articular cartilage, causing its degeneration. Although OA is one of the most prevalent pathologies globally, there are no definitive treatments available. Recently, research has focused on elucidating the complex interplay that takes place between inflammatory processes and epigenetic regulation, showing that histone post-translational modifications (PTMs) can exert a pronounced effect on the expression of OA-related genes. OA chondrocytes enhance the production of interleukin 1β (IL-1β) and interleukin 8 (IL-8), which are epigenetically regulated. These cytokines upregulate the synthesis of matrix metalloproteinases (MMPs) and aggrecanases, which promote the extracellular matrix (ECM) destruction. This motivates the study of histone PTMs to investigate the epigenetic regulation of proinflammatory molecules, but the absence of specific protocols to extract histones from human articular cartilage has complicated this task. The lack of effective methods can be explained by the structural complexity and low cellularity of this tissue, which are responsible for the biomechanical properties that allow the movement of the joint but also complicate histone isolation. Here, we provide a histone extraction procedure specifically adapted for cryopreserved human articular cartilage that can be useful to understand epigenetic regulation in OA and accelerate the search for novel strategies.

## 1. Introduction

### 1.1. The Role of Epigenetics in Osteoarthritis (OA)

Epigenetics is an emerging field that investigates heritable or dynamic changes in gene expression that occur without altering the DNA sequence. DNA methylation and demethylation, histone modifications and regulation by noncoding RNA are the main epigenetic mechanisms involved in the regulation of gene expression [1]. The present study is focused on histones, which are highly basic proteins that constitute the structural unit responsible for eukaryotic DNA packaging, the nucleosome. Histones H2A, H2B, H3 and H4 form histone–histone interactions between their C-terminal domain to comprise a histone octamer onto which the DNA helix is wrapped. On the other hand, histone H1 acts as a linker protein that enables the correct assembly of chromatin [2]. The N-terminal ends of histones often undergo highly dynamic post-translational modifications (PTMs), such as acetylation or methylation [3]. Histone methyltransferases (HMTs) and histones demethylases (HDMTs) catalyse histone methylation and demethylation, respectively, while histone acetyltransferases (HATs) and histone deacetylases (HDACs) make acetylation and deacetylation possible [4]. These PTMs and histone modifiers are able to interact with transcription factors and to alter chromatin conformation, favouring either gene expression activation or repression [5].

In recent years, it has been discovered that epigenetic regulation of gene expression plays a decisive role in the development of certain pathologies. Among them is osteoarthritis (OA), one of the leading causes of disability [6]. OA impairs the osteochondral unit, a biocomposite formed by articular cartilage, subchondral bone and calcified cartilage, causing joint failure and restricting mobility [7]. Contrary to traditional belief, an inflammatory component is present in OA pathogenesis. When the join is injured, chondrocytes respond with an increased production of molecules that are closely related to innate immunity [8]. Foremost among these is interleukin 1β (IL-1β), which is overexpressed in OA cartilage and synovial tissue. IL-1β increases the expression of matrix metalloproteinases (MMPs) and aggrecanases, whose action promotes the destruction of the cartilage extracellular matrix (ECM) and inhibits the synthesis of its major components, such as type II collagen and proteoglycans [5]. Another important element is tumour necrosis factor α (TNF-α), a growth factor that drives the inflammatory signalling cascade by inducing the production of proinflammatory and catabolic molecules [9]. Interleukin 8 (IL-8) is one of the chemokines involved in the inflammatory response initiated by IL-1β and TNF-α, and which has been implicated in proteoglycan loss, MMPs production, chondrocyte hypertrophy and apoptosis, as well as immune system cell entry into synovial tissue [10]. The described proinflammatory cytokines overexpression can occur as a consequence of aberrant epigenetic modulation of their genes. Loss of methylation in specific CpG sites located within *IL1B* and *IL8* promoters is reported to happen in OA chondrocytes after long-term exposure to cytokines [11,12]. Demethylation also occurs at the promoters of *MMP3*, *MMP9* and *ADAMTS4*, leading to an enhanced expression of these ECM degradative enzymes in OA chondrocytes [13,14]. Moreover, it has been shown that the upregulation of inducible nitric oxide synthase (iNOS) in OA is related to the demethylation of an enhancer region, which is one of the binding sites of NF-κB, a transcription factor implicated in several inflammatory diseases [15]. On the contrary, hypermethylation of *COL9A1* in OA chondrocytes results in a decreased production of type IX collagen, a structural molecule that helps to stabilize the collagen network of the cartilage ECM [16].

DNA methylation and histone modifications work together to modulate gene expression. Chromatin regions that present a looser conformation are closely associated with the presence of acetylated histone marks and hypomethylated DNA, which facilitates the access of transcription factors to the DNA. Conversely, the predominance of nonacetylated histones and hypermethylated DNA often results in chromatin compaction, making chromatin inaccessible to the transcription machinery [17]. However, the sequence of events that take place in the existing crosstalk between histone modifications and DNA methylation remains unclear [18]. Considering that age is an important risk factor that increases the likelihood of developing OA [19] and that it can be one of the causes for epigenetic modifications to arise [20], it stands to reason that epigenetics plays an important role in the development of the disease. Therefore, research within this field is necessary and could be useful in the search for new therapeutic strategies to treat OA.

### 1.2. Articular Cartilage: Structure and Composition

Articular cartilage is an avascular, aneural and alymphatic connective tissue that protects diarthrodial joints, allowing a smooth movement with minimal friction and an adequate load transmission to the bone. It is composed of chondrocytes, highly specialized cells that only represent 2% of the total volume of adult cartilage [21]. These synthesize and maintain the components of the extensive ECM in which they are embedded, following a scattered distribution. The ECM accounts for the remaining 98% of tissue volume, mainly composed of water (60–80%) and structural molecules (20–40%), including collagens, proteoglycans, noncollagenous proteins and glycoproteins [22]. One of the largest groups of structural molecules present in articular cartilage is collagen, which varies in thickness and distribution in each layer of tissue. Its triple-helix molecular structure is partly responsible for the tensile properties of cartilage [23]. Another type of molecule that plays a pivotal role in the ECM is proteoglycans such as aggrecan. These tend to form negatively charged aggregates when a noncovalent association between them and hyaluronic acid takes place [24]. Anions present in proteoglycans attract cations that are dissolved in the aqueous phase of the ECM, establishing electrostatic forces that, in turn, attract water [25]. As a result, the aggrecan matrix expands and the collagen network prevents it from swelling, generating a high osmotic pressure within the tissue [26]. This phenomenon provides stiffness and compression resistance, stabilizing the architecture of the ECM. In normal (N) cartilage (Figure 1A, magnified in Figure 1a), chondrocytes control the balance between anabolism and catabolism of all the mentioned constituents, which are organized into three distinctive zones that give the tissue its mechanical properties [23] (Table 1).

In OA cartilage, this equilibrium is compromised and an increased production of proinflammatory cytokines and degradative enzymes is induced [7]. In addition, a decrease in the number of chondrocytes occurs, and the remaining cells are incapable of remodelling the ECM. In an attempt to compensate the loss of structural molecules in the matrix, some chondrocytes replicate and form groups of approximately 50 cells that are known as “chondrocyte clones” (Figure 1B, magnified in Figure 1b). However, cartilage structure remains damaged [27]. Cells comprising these clusters have a repertoire of genes whose epigenetic pattern is aberrant and which they have inherited during mitosis. For this reason, OA chondrocytes neo-express a battery of degradative enzymes that N chondrocytes cannot express due to functional epigenetic silencing [17].

### 1.3. Lack of Articular-Specific Histone Extraction Methods

The structural complexity and biochemical properties of cartilage together with its low cellularity complicate the performance of certain in vitro procedures, such as nucleic acid or protein extraction. After having tested multiple protocols with the purpose of extracting histones from cryopreserved human cartilage, we realize that the homogenization process becomes a crucial phase in the whole extraction procedure, as it conditions the accessibility of all the reagents to the chondrocytes. The study of histones could be relevant to elucidate the molecular patterns that regulate OA progression. Thus, we have adapted the method provided by Rivera-Casas and collaborators [28] for characterizing chromatin-associated proteins in bivalve molluscs to articular cartilage necessities, emphasizing the relevance of the homogenization steps in this tissue.

## 2. Methods and Results

### 2.1. Starting Material

The samples needed to carry out this study were obtained from the collection of samples for the investigation of Rheumatic Diseases of Dr. Francisco J. Blanco García, from Xerencia de Xestion Integrada de A Coruña (XXIAC) and the Institute of Biomedical Research of A Coruña (INIBIC). This collection was registered in the National Registry of Biobanks, with registration code: C.0000424 and approved by the Ethics Committee of Galicia with registration code: 2013/107. When choosing the starting material, we recommend using ~500 mg up to 1 g of cryopreserved cartilage to obtain reasonable quantities of histones (Table 2). It must be borne in mind that extraction yields from N cartilage are usually lower than yields from OA cartilage (Table 2), given that chondrocyte clones can be present in the latter but not in the former.

### 2.2. Pulverization of Cryopreserved Articular Cartilage Samples

Cartilage requires a pulverization step to achieve an effective homogenization. This is preferably performed using a cylindrical flat-bottomed mortar and a pestle, both made of a resistant material such as steel. It is important that the pestle fits the mortar perfectly to avoid its movement, as pulverization will be carried out by hammering the sample. It is crucial to take into consideration that to prevent protein degradation, all the materials and reagents that will be in contact with the sample should be precooled. Prior to pulverization, place the mortar and the pestle in a foil-covered box, pour liquid nitrogen into it and wait until the tools are cold.Introduce the cryogenic tube containing the sample in liquid nitrogen for approximately 5 min. Tissue must be completely frozen, as thawing will hamper its fragmentation and promote protein degradation.Take the precooled mortar out of the box and place on it a flat surface.Remove the sample from liquid nitrogen and set it in the centre of the mortar.Pulverize the tissue by firmly hitting the pestle against the mortar.Repeat the process until cartilage acquires a powdery consistency.Separate a part of the resulting material with a 5 mm lab spoon into a 1.5 mL microcentrifuge tube. This will be used to obtain the total protein extract of tissue samples.Transfer the resulting material to an ice-cold 15 mL tube. Although it is not necessary to use 15 mL tubes, we propose this size because it offers enough space to allow an appropriate homogenization, as the reagents will reach the totality of the sample when vortexing. From now on, all steps must be performed on ice unless otherwise specified.

### 2.3. Tissue Homogenization

Homogenization will be achieved using two homogenization buffers (Buffer A and Buffer B), which were developed by Rivera-Casas and colleagues [28]. Considering that protease content in OA cartilage is noteworthy, homogenization buffers can be supplemented with different protease inhibitors before use. According to Rumbaugh and Miller’s guidelines [29], we supplemented both buffers as detailed in Table 3.
Add 2 mL of Buffer A to the 15 mL tube and vortex thoroughly.Incubate the sample on ice for 10 min (this will allow Buffer A to disrupt cell membranes).Vortex the sample, and centrifuge at 4000× *g* for 10 min at 4 °C. Transfer the supernatant containing the cytosolic fraction (cytosolic fraction 1, CF1) to a fresh 2 mL centrifuge tube and maintain it on ice.Due to the low cellularity and ECM abundance in articular cartilage, we recommend performing this homogenization step twice to obtain higher extraction yields. Again, transfer the supernatant containing the cytosolic fraction (cytosolic fraction 2, CF2) to a fresh 2 mL centrifuge tube and keep it on ice. The volume of Buffer A used to homogenize cartilage is considerable, so cytosolic proteins contained in CF1 and CF2 should be concentrated before using them in other assays. Amicon^®^ Ultra-2 Centrifugal Filter Units for concentration and purification of biological solutions (Merck Millipore) can be used for this purpose following manufacturer’s instructions.Add 2 mL of Buffer B.Vortex the sample to obtain a viscous homogenate and incubate it on ice for 10 min.Vortex the sample, centrifuge at 4000× *g* for 10 min at 4 °C and discard the supernatant.Centrifuge at 4000× *g* for 10 min at 4 °C without adding any buffer to completely remove the remaining Buffer B that could be infiltrated between the tissue fragments of the pellet. Again, discard the supernatant. (Note: Do not expect to obtain a “conventional” pellet, since at this point rest of the ECM will remain, forming a granular pellet.)

### 2.4. Histone Extraction

Add 1 mL of 1.2 M H_2_SO_4_ to solubilize histones and vortex the sample.Centrifuge at 5500× *g* for 15 min at 4 °C.Transfer the supernatant to a 2 mL microcentrifuge tube.Add 1 mL of cold acetone (−20 °C).Invert the tube a few times to determine whether a whitish precipitate has formed, meaning that histone precipitation is occurring. If the starting material is scarce, you may not see the reaction immediately. However, after an overnight incubation of the samples at −20 °C, a cloudy precipitate should be visible at the bottom of the tube.Centrifuge the samples at 10,000× *g* for 10 min at 4 °C and discard the supernatant.Subject the pellet to a second wash with 1 mL of cold acetone (−20 °C).Centrifuge at 10,000× *g* for 10 min at 4 °C and discard the supernatant.Let the pellet dry for ~30 min, checking that the acetone has been totally evaporated before resuspension. If not, carefully remove it by pipetting or clean the walls of the tube with tissue paper, avoiding touching the pellet.U/T buffer (25 mM ammonium bicarbonate, 6 M urea, 2 M thiourea) facilitates resuspension, so we recommend its use instead of dH_2_O. Add an appropriate volume of U/T buffer to resuspend the pellet, taking care not to introduce any air bubbles with the pipette. Scrape the wall of the tube with the pipette tip to detach the pellet and patiently dissolve it by vortexing. Verify that no clumps remain and that the whole precipitate has been dissolved.Protein concentration can be quantified with the Bradford assay. Histone extracts obtained with this protocol cannot be quantified with Pierce^TM^ BCA Protein Assay Kit (Thermo Fisher Scientific) due to the incompatibility with the U/T buffer.Store histone extracts at −80 °C or prepare aliquots containing equal amounts of protein in a 1:10 mixture of β-mercaptoethanol and 4× Laemmli Sample Buffer (Bio-Rad) for further procedures.

### 2.5. Total Protein Extract Obtention

Add two zirconium oxide grinding balls to each 1.5 mL microcentrifuge tube.Add 200 µL of U/S buffer (6 M urea, 2% SDS).Vigorously vortex the samples for 2 h at RT.Sonicate the samples for 5 min at RT.Centrifuge at full speed for 1 min at RT.Transfer the supernatant to a fresh 1.5 mL microcentrifuge tube.Centrifuge at full speed for 15 min at RT.To ensure the complete removal of insoluble material, transfer the supernatant to a fresh 1.5 mL microcentrifuge tube and discard the pellet.Protein concentration can be quantified with the Bradford assay.Store total protein extracts at −20 °C or prepare aliquots containing equal amounts of protein in a 1:10 mixture of β-mercaptoethanol and 4× Laemmli Sample Buffer (Bio-Rad) for further procedures.

### 2.6. Electrophoretic Separation of Proteins on Polyacrylamide Gels

Vortex protein samples before denaturation.Boil the samples for 5 min at 95 °C and immediately put them on ice.Place the polyacrylamide gels in a vertical electrophoresis system and fill the electrophoresis tank with fresh 10% running buffer (Trizma base, glycine, 20% SDS, dH_2_O).Vortex the samples again.Load equal sample volumes into the wells of the gel. You can prepare your own gels as long as its polyacrylamide percentage is set between 15–20%, since the molecular weight of histones requires a resolving gel with small pore size (see recipe in Table 4). For better results in Western blotting, we recommend using 4–20% Mini-PROTEAN^®^ TGX^TM^ Precast Protein Gels (Bio-Rad). A comparative example of both types of gels performance in Western blotting is shown in Figure 2.Run the SDS-PAGE electrophoresis at 70 V for 20 min or until proteins have reached the stacking gel limit. Then, raise the voltage to 100 V for 1 h 20 min. For Western blotting procedures, soak nitrocellulose membranes of 0.2 µm pore size in transfer buffer (Table 5) with constant shaking while the electrophoresis is running. Although 0.45 µm pore size membranes can be used (whether PVDF or nitrocellulose), we recommend using 0.2 µm nitrocellulose membranes because of their high capacity to retain lower molecular weight proteins. This is especially useful for the study of smaller histones such as H4.

### 2.7. Silver Staining

In order to check the purity of histone extracts, it is recommended that a silver staining of the polyacrylamide gel be performed after completion of electrophoresis. As described in [28], it is worth noting that SDS gels do not have enough resolution to clearly distinguish histone variants and histones with different PTMs from their canonical forms. For this aim, acetic-urea-triton (AUT) and acid-urea (AU) gels are more suitable since they separate proteins according to their effective charge instead of molecular weight [28]. However, we used a 4–20% Mini-PROTEAN^®^ TGX^TM^ Precast Protein Gels (Bio-Rad) to verify that the extraction technique yields a pure histone extract compared to the total extract and cytosolic fractions using a silver staining protocol modified from Blum and collaborators (1987) [30] (Figure 3).
Remove the gel from the cassette and soak it in 50 mL of fixative solution (40% ethanol, 10% glacial acetic acid). Fix the gel for 45 min at RT with smooth constant shaking.Wash the gel 2 times × 10 min with dH_2_O with smooth, constant shaking.Sensitize the gel in 0.02% sodium thiosulfate solution for 1 min.Wash the gel 2 times × 1 min with dH_2_O.Impregnate the gel with 0.2% silver nitrate and 0.075% formalin solution for 45 min with smooth, constant shaking.Wash the gel 2 times × 10 s with dH_2_O.Detect proteins in the gel by exposing the membrane to 3% sodium carbonate, 12.5 mg/L sodium thiosulfate and 0.025% formalin solution. Shake the gel vigorously for 3–10 min or until a brown precipitate is visible, as shown in Figure 3. Do not extend this step for more than 15 min.Stop the image development reaction by immersing the gel in 3% Trizma base and 10% glacial acetic acid solution for 30 min with smooth constant shaking.The gel can be conserved in dH_2_O at 4 °C.

### 2.8. Western Blot

When the electrophoresis dye front has reached the bottom of the gel, proceed to transfer the gel to a nitrocellulose membrane using fresh cold transfer buffer (Table 5). Given that the transfer process generates a considerable amount of heat, place the tank into a box filled with ice. Additionally, a cold accumulator can be introduced into the tank next to the transfer cassette. Set the voltage to 100 V for 1 h 30 min.When the transfer has finished, immerse the membranes in 5% blocking solution (Table 5) for 1 h at RT with constant shaking.Incubate them with specific primary antibodies diluted in 5% blocking solution in a rotator at 4 °C overnight or as long as necessary.The following day, wash the membranes 3 times × 10 min at RT with 10% TBST (Table 5) in constant shaking.Incubate the membranes with a secondary antibody for 1 h at RT with constant shaking.Repeat the 10% TBST washes.Detection of the protein signal can be performed using an enhanced chemiluminescent (ECL) system, for instance, Immobilon Classico Western HRP substrate (Merck Millipore). An example with two histone modifications is shown in Figure 4.

### 2.9. Membrane Stripping

When investigating multiple proteins with the same molecular weight, removing previously incubated primary and secondary antibodies of a blot can be useful. For that reason, we include the following protocol:Immerse the membranes in stripping solution (0.5 M Tris-HCl, 10% SDS, 100 mM β-mercaptoethanol, dH_2_O) for 45 min at 50 °C with constant shaking.Wash the membranes in dH_2_O for 2 h at RT with constant shaking. Change the dH_2_O regularly to remove all traces of stripping solution.Wash the membranes in 10% TBST for 30 min at RT with constant shaking. At this point, it is advisable to check if protein signal is completely removed.Soak the membranes in 5% blocking solution for 1 h at RT with constant shaking.Follow steps 3–7 from Section 2.8.

## 3. Conclusions

The present protocol constitutes a novel strategy that makes it possible to extract histones from articular cartilage without having to resort to chondrocyte isolation. In this technique, enzymatic digestion of the tissue is performed, first with trypsin and then with collagenase. Since the effects of enzymatic treatments on histone PTMs are unknown, we believe that being able to use cartilage as a starting material is an inherent advantage of our protocol. Moreover, the procedure is inexpensive compared to commercial kits, which may help to save costs when working with a large number of samples.

Histone extraction from human articular cartilage can be challenging if certain methodological aspects are not considered, which prove to be key points for successful isolation. As discussed above, complete pulverization of cartilage is essential for a satisfactory extraction, as this increases the surface area on which the reagents act. Homogenization is also a step of significant relevance, where selection of buffers and protease inhibitors must be appropriate for the type of tissue we are working with. Nevertheless, having sufficient space for the reagents to reach the chondrocytes and mixing them well by vortexing is probably most important. Another point to note is that the concentration of the acid used to solubilize histones should not be too low, at least in the particular case of articular cartilage. After several tests, our group found that acid concentrations widely used to extract histones from other types of tissues (0.2 M, 0.4 M, 0.6 M) turned out to be insufficient to obtain good yields in cartilage.

The pathogenesis of OA encompasses multiple molecular mechanisms that work in an interconnected manner, many of which are epigenetically regulated. It is therefore imperative to understand how this modulation works and what its implications are in the inflammatory processes that take place in OA joints. Recently, it has been discovered that histone PTMs may have a major influence on the metabolic imbalance that occurs in chondrocytes. Hence, their study could allow the search for more specific therapeutic targets or even the design of new strategies to reduce joint degradation or prevent OA. The aim of the present protocol for histone extraction from human articular cartilage is to facilitate this task and serve as an aid to understanding OA, an increasingly prevalent disease.

## Figures and Tables

**Figure 1 ijms-23-03355-f001:**
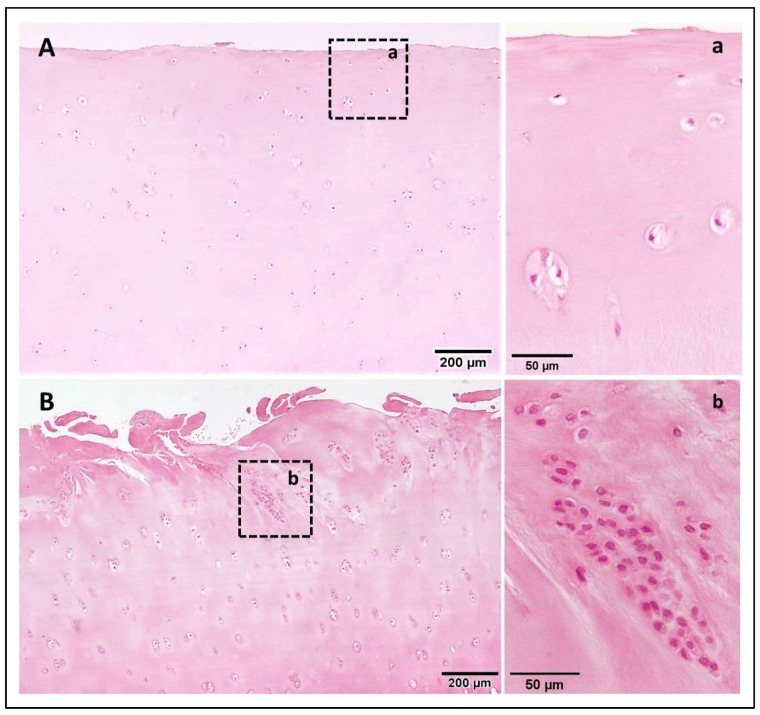
Haematoxylin and eosin staining of human N and OA cartilage. (**A**) Longitudinal section of N cartilage. (**a**) Detail of (**A**). (**B**) longitudinal section of OA cartilage. (**b**) Detail of (**B**) showing an example of chondrocyte clones.

**Figure 2 ijms-23-03355-f002:**
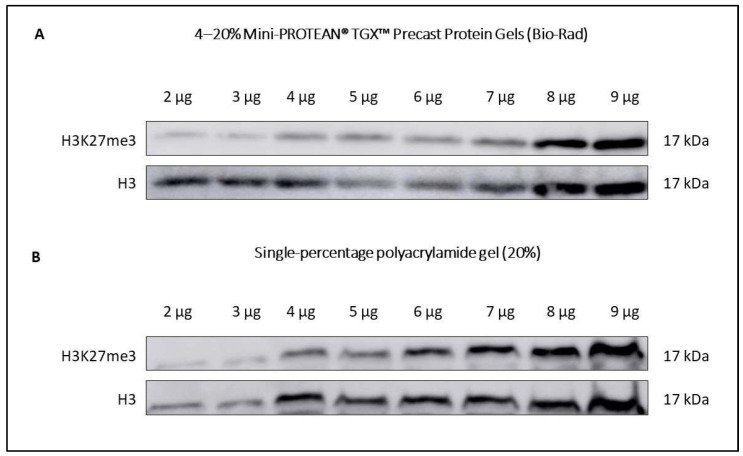
Analysis of different protein concentrations to study H3K27me3 in OA1 cartilage by Western blotting. The weight of the starting material was 1.20 g. (**A**) SDS-PAGE electrophoresis was performed using 4–20% Mini-PROTEAN**^®^** TGX^TM^ Precast Protein Gels (Bio-Rad). (**B**) SDS-PAGE electrophoresis was performed using a single-percentage polyacrylamide gel (20%). Both gels were transferred to 0.2 µm nitrocellulose membranes (Bio-Rad) to incubate them with primary antibodies against H3K27me3 (#9733, Cell Signaling Technology, 1:1000) overnight at 4 °C and H3 (#14269, Cell Signaling Technology, 1:10,000) for 5 min (**A**) and 1 h (**B**) at room temperature (RT). Secondary antibodies Rabbit IgG HRP Linked Whole Ab (NA934, 1:2000) and Mouse IgG HRP Linked Whole Ab (NA931, 1:2500) were used, respectively.

**Figure 3 ijms-23-03355-f003:**
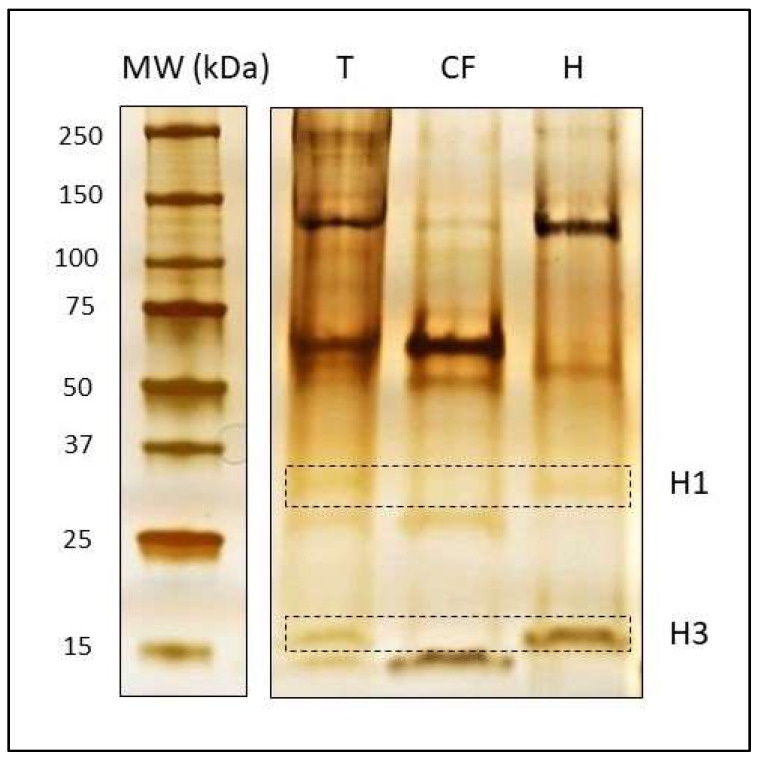
Silver staining of proteins after electrophoretic separation on 4–20% polyacrylamide gel. (MW) Molecular weight marker. (T) Total protein extract. (CF) Cytosolic fraction (concentrated CF1 + CF2). (H) Histone extract. Two micrograms of protein were loaded in each lane. All samples correspond to OA3.

**Figure 4 ijms-23-03355-f004:**
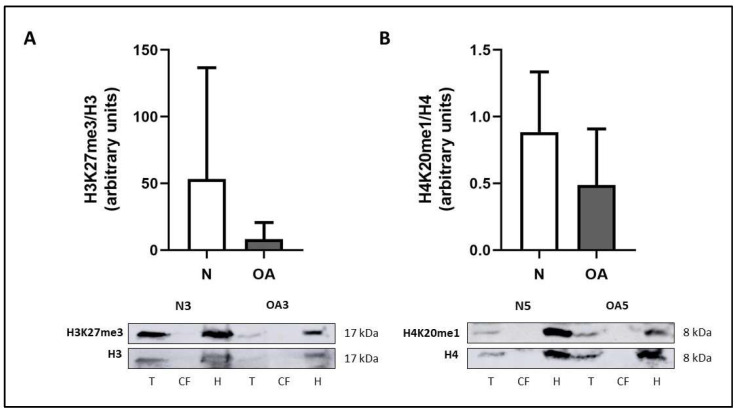
Comparison of histone marks H3K27me3 and H4K20me1 in OA and N cartilage by Western blotting. Proteins from total extract (T), cytosolic fraction (CF) and histone extract (H) were loaded. Ten micrograms of protein were loaded in each lane. (**A**) H3K27me3 (#9733, Cell Signaling Technology, 1:1000) expression relative to H3 (#14269, Cell Signaling Technology, 1:10,000) in OA and N cartilage. (**B**) H4K20me1 (39027, Active Motif, 1:1000) expression relative to H4 (61521, Active Motif, 1:10,000) in OA and N cartilage. Secondary antibody Rabbit IgG HRP Linked Whole Ab (NA934, 1:2000) and Mouse IgG HRP Linked Whole Ab (NA931, 1:2500) were used.

**Table 1 ijms-23-03355-t001:** Structural characteristics of normal osteochondral unit.

		ECM			Chondrocytes	
Zone	Function	Collagen		Proteoglycans	Shape	Abundance
		Fibres	Distribution	Abundance		
Superficial	Protecting the underlying layers and providing elasticity	Packed	Parallel	Low	Flattened	Relatively high
Middle	First defence line against compressive forces	Thick	Oblique	High	Spherical	Low
Deep	Resistance against compressive forces	Very thick	Perpendicular	Very high	Spherical, columnar arrangement	Relatively high
Calcified	Anchoring of collagen fibres to the bone	Anchored to the subchondral bone		-	Hypertrophied	

**Table 2 ijms-23-03355-t002:** Protein concentration yielded after histone extraction in relation to the weight of the sample. Quantification methods (bicinchoninic acid (BCA) and Bradford) and resuspension solvents (H_2_O and urea/thiourea (U/T) buffer) used in each case are indicated below.

Sample	Weight	Protein Yield	Quantification Method	Resuspension Solvent
OA1	1.20 g	5.30 µg/µL	BCA	H_2_O
OA2	350 mg	1.94 µg/µL	BCA	H_2_O
OA3	556 mg	4.04 µg/µL	Bradford	U/T
OA4	797 mg	2.86 µg/µL	Bradford	U/T
OA5	952 mg	2.60 µg/µL	Bradford	U/T
N1	1.20 g	2.39 µg/µL	BCA	H_2_O
N2	140 mg	0.82 µg/µL	BCA	H_2_O
N3	1.64 g	1.24 µg/µL	Bradford	U/T
N4	1.98 g	1.29 µg/µL	Bradford	U/T
N5	1.28 g	0.67 µg/µL	Bradford	U/T

**Table 3 ijms-23-03355-t003:** Homogenization buffers composition and supplementation.

Buffer A	Buffer B
0.15 M NaCl	0.1 M KCl
10 mM Tris-HCl pH = 7.5	50 mM Tris-HCl pH = 7.5
0.5% Triton X-100	1 mM MgCl_2_
2 mM sodium butyrate	
1 mM sodium orthovanadate	
0.5 mM PMSF	
1× protease inhibitor cocktail (Sigma)	

**Table 4 ijms-23-03355-t004:** Recipes of single-percentage polyacrylamide gels. Volumes were calculated following Bio-Rad instructions.

		Stacking Gel	Running Gel	
		4%	15%	20%
dH_2_O		4.8 mL	4.7 mL	1.4 mL
0.5 M Tris-HCl pH = 6.8		2.0 mL	–	–
1.5 M Tris-HCl pH = 8.8		–	5.0 mL	5.0 mL
30% Acrylamide/Bis		1.1 mL	10 mL	13.3 mL
10% Sodium dodecyl sulfate (SDS)		80 µL	200 µL	200 µL
10% Ammonium persulfate (APS)		40 µL	100 µL	100 µL
Tetramethylethylenediamine (TEMED)		8.0 µL	10 µL	10 µL

**Table 5 ijms-23-03355-t005:** Composition and storage of Western blot reagents.

Reagent	Composition	Storage
Transfer buffer	100% running buffer	4 °C until use
	100% ethanol	
	dH_2_O	
100% wash buffer (TBST)	Trizma base	RT
	NaCl	
	Tween-20	
	dH_2_O	
5% blocking solution	Bovine Serum Albumin (BSA)	4 °C
	10% TBST	

## Data Availability

The data are available from the corresponding author upon request.

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
