# Peer review of "Histone Extraction from Human Articular Cartilage for the Study of Epigenetic Regulation in Osteoarthritis"

_ijms, 2022, doi:10.3390/ijms23063355_

Round 1

Reviewer 1 Report

This article is an attempt to present a developed protocol of histone extraction from human articular cartilage for osteoarthritis studies. The main problem with this study is a little number of examined cartilage samples (2N and 2OA), eg. when age is an important risk factor for OA (see 90th row). It would be very informative to present data on patients with osteoarthritis. Probably ethical issues for samples deposited in the biobank should be included in the manuscript. Moreover, H3 methylation provide single feature of epigenetic regulation, regarding to histone modification. To add, protocol for H3K27me3 and H3K9me3 was not provide sufficient data on the epigenetic regulation of proinflammatory molecules.

Author Response

Based on the helpful suggestions of all reviewers, we have implemented several improvements to the protocol that help to take full advantage of the described technique. The modifications added to the histone extraction method include:
-    An extension of Table 2, showing the number of samples used (now 5 OA and 5N), the weight of starting material, the protein concentration obtained after histone extraction, the quantification method used, and the resuspension solvent used in each case.
-    A brief protocol for obtaining total articular cartilage protein extract from pulverized material (see subsections 2.2. Pulverization of Cryopreserved Articular Cartilage Samples and 2.5. Total protein extract obtention in the manuscript).
-    A way to take advantage of the cytosolic fractions obtained in the first steps of the extraction for later use in other techniques (see 2.3. Tissue Homogenization in the manuscript).
-    The use of U/T buffer as an alternative to using water to resuspend histone extracts, which facilitates the resuspension process (see 2.4. Histone extraction in the manuscript). The U/T buffer has the minor disadvantage that extracts resuspended in it cannot be quantified by the BCA assay, as mentioned in lines 240-242. However, this is quickly overcome by using one of the best known and most accessible protein quantification assays: the Bradford method.
-    The use of 0.2 µm nitrocellulose membranes instead of 0.45 µm because of their high capacity to retain lower molecular weight proteins, such as histone H4.
-    An evaluation of histone extract purity in comparison with total protein extract and cytosolic fraction by silver staining the polyacrylamide gel (see 2.7. Silver staining in the manuscript).
-    A brief protocol to perform membrane stripping that removes previously incubated primary and secondary antibodies of a blot (see 2.9. Membrane stripping in the manuscript. This allows to study proteins with the same molecular weight in the same membrane.
Reviewer 1:
We welcome the comment of reviewer 1 and thank reviewer 1 for his/her very constructive criticism on our manuscript. The following address the concerns of reviewer 1:
1.  What is the advantage of your method over commercial kits, what are the differences? Please include this in conclusions.
We thank the reviewer for this interesting question; one obvious answer is costs saving when you are planning to analyse many samples, nevertheless we optimized this protocol because cartilage is a difficult sample type to work with. After contacting specific companies working on epigenetics (i.e., Active Motif) we were advised to isolate chondrocytes before proceeding to histone extraction. This is a protocol widely used for us and others, but we wondered if enzymatic digestion with trypsin first and collagenase afterwards will modify post-translational modifications (PTMs) in histones, for that reason we decided to use cartilage as starting material. Also, this is indeed a comment that we always get in any paper revision, it is always better to work with cells when they are in their native environment. Furthermore, the effect of enzymatic digestion on PTMs is an interesting point that we are planning to study in the near future.
Please note we have accordingly added this statement to the conclusions section.
2. Can all types of histones be isolated using this method or just the one mentioned in the protocol? Did you check this method for other histone types?
All types of histones can be isolated using this method because it is based in the basic charge of these proteins. Initially, we had only used the technique to detect histone H3, the most studied canonical histone due to the large variety of residues that are susceptible to suffer PTMs. Based on the reviewers' suggestions, we purchased specific antibodies against H4K20me1 and H4 to show in the updated version of the manuscript that it is possible to detect these marks with our extraction technique. As shown in Figure 3 canonical histones H1 and H3 are detected and H4 is shown in Figure 4. In order to detect histone variants (i.e., H2A.X and H2A.Z) different type of gels should be used as described in Rivera-Casas et al. (2017) and mentioned in lines 312-317, but it is outside of the scope of this paper.
3. Can we use this protocol for other hard tissues?
Although we did not try with other tissues, we anticipate standard isolation protocol using diluted acids can be easily implemented. However, extraction procedures still require modifications to account for specific particularities, notably in the case of nuclei isolation.  The low cellularity of articular cartilage and the ECM complexity make the isolation difficult, other tissues should be more kind to histone extraction. In the case of bone, a previous de-calcification step should be done first and then proceed with this protocol.
4. Did you check the purity of histone extracts by other methods than western blot? For example, HPLC?
We thank the reviewer for this comment. We did not have the possibility to check the purity with HPLC considering time limitation, but we have compared in a silver-stained SDS-PAGE gel all three protein fractions: total, cytosolic and histones. We show how histones appear in this last fraction and not in the cytosolic, both in the silver-stained gel (Figure 3) and in the blots incubated with H3K27me3 and H4K20me1 (Figure 4), showing the purity of the extracts.
5. How do you explain double bands on blots of H3 in non-published material?
We thank the reviewer for this helpful comment. We had to modify our standard protocol with 0.45 µm pore size polyvinylidene difluoride (PVDF) membranes to 0.2 µm pore size nitrocellulose membranes to be able to retain these small proteins. Detection signals were higher than expected, we have now decreased to 5 min incubation times only instead of 1 hour for H3 and H4 in order to get a single discrete band. However, we were only able to do this in the blot corresponding to 4–20% Mini-PROTEAN® TGX™ Precast Protein Gels (Bio-Rad) shown in Figure 2 and not in the blot corresponding to single-percentage polyacrylamide gel (20%).
We have accordingly modified the figures and attached the complete blots.

Reviewer 2 Report

The manuscript entitled Histone Extraction from Human Articular Cartilage for the Study of Epigenetic Regulation in Osteoarthritis by Carmen Núñez-Carro and collegues, present enhanced protocol of histone extraction from articular cartilage. It is problematic tissue to work with therefore presented manuscript helps to readers overcome difficulties with histone extraction.  The subject and aim of this study would be of interest for readers of JIMS, however please answer some questions:

- What is the advantage of your method over commercial kits, what are the differences? Please include its in conclusions

- Can all types of histones be isolated using this method or just the one mentioned in the protocol? Did you check this method fot other histone types?

- Can we use this protocol for other hard tissues?

-Did you check the purity of histone extracts by other methods than western blott? For example HPLC?

-How do you explain double bands on blots of H3 in non-published material?

Author Response

Based on the helpful suggestions of all reviewers, we have implemented several improvements to the protocol that help to take full advantage of the described technique. The modifications added to the histone extraction method include:

  • An extension of Table 2, showing the number of samples used (now 5 OA and 5N), the weight of starting material, the protein concentration obtained after histone extraction, the quantification method used, and the resuspension solvent used in each case.
  • A brief protocol for obtaining total articular cartilage protein extract from pulverized material (see subsections 2. Pulverization of Cryopreserved Articular Cartilage Samples and 2.5. Total protein extract obtention in the manuscript).
  • A way to take advantage of the cytosolic fractions obtained in the first steps of the extraction for later use in other techniques (see 3. Tissue Homogenization in the manuscript).
  • The use of U/T buffer as an alternative to using water to resuspend histone extracts, which facilitates the resuspension process (see 4. Histone extraction in the manuscript). The U/T buffer has the minor disadvantage that extracts resuspended in it cannot be quantified by the BCA assay, as mentioned in lines 240-242. However, this is quickly overcome by using one of the best known and most accessible protein quantification assays: the Bradford method.
  • The use of 0.2 µm nitrocellulose membranes instead of 0.45 µm because of their high capacity to retain lower molecular weight proteins, such as histone H4.
  • An evaluation of histone extract purity in comparison with total protein extract and cytosolic fraction by silver staining the polyacrylamide gel (see 7. Silver staining in the manuscript).
  • A brief protocol to perform membrane stripping that removes previously incubated primary and secondary antibodies of a blot (see 9. Membrane stripping in the manuscript. This allows to study proteins with the same molecular weight in the same membrane.

Reviewer 2:

We welcome the comment of reviewer 2 and thank reviewer 2 for his/her very constructive criticism on our manuscript. The following address the concerns of reviewer 2.

  1. The main problem with this study is a little number of examined cartilage samples (2N and 2OA), eg. when age is an important risk factor for OA (see 90throw). It would be very informative to present data on patients with osteoarthritis.

We thank the reviewer for this helpful comment. We have now added more samples (which are of similar ages) and, although we have not found significant differences between both types of samples (N and OA), there is an evident trend that can be observed in Figure 4. However, in order to get significant information about age we will need a substantial number of samples. This is a running study at the moment in the lab. Also, the objective of this manuscript was to set up a protocol for histone extraction, never done as far as we know.

  1. Probably ethical issues for samples deposited in the biobank should be included in the manuscript.

We thank the reviewer for this clarification. We have now added to the manuscript the following statement (lines 149-154):

The samples needed to carry out this study were obtained from the collection of samples for the investigation of Rheumatic Diseases of Dr. Francisco J. Blanco García, from Xerencia de Xestion Integrada de A Coruña (XXIAC) and the Institute of Biomedical Research of A Coruña (INIBIC). This collection was registered in the National Registry of Biobanks, with registration code: C.0000424 and approved by the Ethics Committee of Galicia with registration code: 2013/107.

  1. Moreover, H3 methylation provide single feature of epigenetic regulation, regarding to histone modification. To add, protocol for H3K27me3 and H3K9me3 was not provide sufficient data on the epigenetic regulation of proinflammatory molecules.

We thank the reviewer for this interesting point. The aim of this article is to provide a protocol that allows the extraction of histones from articular cartilage, which in turn will allow the study of a wide range of histone PTMs and the implications they have in the regulation of inflammatory processes. In fact, we are currently working on this matter in the lab. There is evidence that the proinflammatory cytokine IL-1β, involved in OA, can regulate the expression levels of EZH2, a methyltransferase that catalyses H3K27 trimethylation (Chen et al. (2017), doi: 10.1038/srep29176). We have included H4K20me1 as a subject of study instead of H3K9me3 due to time limitation, but we will continue to investigate this epigenetic mark in the future. Although there is no information about H4K20me1 implications in OA, it is known that H4K20me1 is highly abundant in transcriptionally active chromatin regions, unlike its di- and tri- methylated versions (Barter et al. (2012), doi: https://doi.org/10.1016/j.joca.2011.12.012).Therefore, it is interesting for us to investigate if H4K20me1 exerts important effects on OA pathogenesis.

Reviewer 3 Report

The clearly written protocol “Histone Extraction from Human Articular Cartilage for the Study of Epigenetic Regulation in Osteoarthritis” describes precisely the steps for an optimal isolation of histones from cartilage. The different steps are clearly described so that other researchers should be able to follow the protocol for their own projects.

Figure 2: Which OA sample is shown? OA1 or OA 2? The information is missing.

However, I have a few more general questions/comments:

What is the conclusion from figure 3 because there is no obvious difference between OA1 and N1?

What is the difference between N1 and N2? Individual differences? What would be then the correlation to OA?

Author Response

Based on the helpful suggestions of all reviewers, we have implemented several improvements to the protocol that help to take full advantage of the described technique. The modifications added to the histone extraction method include:

  • An extension of Table 2, showing the number of samples used (now 5 OA and 5N), the weight of starting material, the protein concentration obtained after histone extraction, the quantification method used, and the resuspension solvent used in each case.
  • A brief protocol for obtaining total articular cartilage protein extract from pulverized material (see subsections 2. Pulverization of Cryopreserved Articular Cartilage Samples and 2.5. Total protein extract obtention in the manuscript).
  • A way to take advantage of the cytosolic fractions obtained in the first steps of the extraction for later use in other techniques (see 3. Tissue Homogenization in the manuscript).
  • The use of U/T buffer as an alternative to using water to resuspend histone extracts, which facilitates the resuspension process (see 4. Histone extraction in the manuscript). The U/T buffer has the minor disadvantage that extracts resuspended in it cannot be quantified by the BCA assay, as mentioned in lines 240-242. However, this is quickly overcome by using one of the best known and most accessible protein quantification assays: the Bradford method.
  • The use of 0.2 µm nitrocellulose membranes instead of 0.45 µm because of their high capacity to retain lower molecular weight proteins, such as histone H4.
  • An evaluation of histone extract purity in comparison with total protein extract and cytosolic fraction by silver staining the polyacrylamide gel (see 7. Silver staining in the manuscript).
  • A brief protocol to perform membrane stripping that removes previously incubated primary and secondary antibodies of a blot (see 9. Membrane stripping in the manuscript. This allows to study proteins with the same molecular weight in the same membrane.

Reviewer 3:

We thank reviewer 3 for his/her helpful comments. We have addressed the concerns of reviewer 3 as follows:

  1. Figure 2: Which OA sample is shown? OA1 or OA 2? The information is missing.

We thank the reviewer for this comment. The sample shown in Figure 2 is OA1, this information has been added to the figure legend.

  1. What is the conclusion from figure 3 because there is no obvious difference between OA1 and N1?

We thank reviewer 3 for this helpful comment. Also following reviewer 2 comments, we have now added more samples, both N and OA and differences between them are cleared now. As shown in Figure 4, H3K27me3 and H4K20me1 expression is higher in N compared to OA samples.

  1. What is the difference between N1 and N2? Individual differences? What would be then the correlation to OA?

We do apologise if differences between samples (amount of cartilage) were not clearly described along the text. To avoid this confusion now we have removed Figure 3 and expanded the number of N and OA samples to check if there are differences in H3K27me3 and H4K20me1 expression levels between both types of samples. We found that both epigenetic marks expression levels are higher in N samples (shown in Figure 4). We are currently working on elucidating whether if low expression levels in OA are somehow correlated to age and/or to K-L grade of OA (Kellgren–Lawrence scoring system).